# The Multiple Cellular Roles of SMUG1 in Genome Maintenance and Cancer

**DOI:** 10.3390/ijms22041981

**Published:** 2021-02-17

**Authors:** Sripriya Raja, Bennett Van Houten

**Affiliations:** 1Molecular Pharmacology Graduate Program, School of Medicine, University of Pittsburgh, Pittsburgh, PA 15213, USA; srr66@pitt.edu; 2UPMC Hillman Cancer Center, University of Pittsburgh, Pittsburgh, PA 15213, USA; 3Department of Pharmacology and Chemical Biology, School of Medicine, University of Pittsburgh, Pittsburgh, PA 15213, USA

**Keywords:** DNA damage, base excision repair, SMUG1, 5-hmdU, cancer

## Abstract

Single-strand selective monofunctional uracil DNA glycosylase 1 (SMUG1) works to remove uracil and certain oxidized bases from DNA during base excision repair (BER). This review provides a historical characterization of SMUG1 and 5-hydroxymethyl-2′-deoxyuridine (5-hmdU) one important substrate of this enzyme. Biochemical and structural analyses provide remarkable insight into the mechanism of this glycosylase: SMUG1 has a unique helical wedge that influences damage recognition during repair. Rodent studies suggest that, while SMUG1 shares substrate specificity with another uracil glycosylase UNG2, loss of SMUG1 can have unique cellular phenotypes. This review highlights the multiple roles SMUG1 may play in preserving genome stability, and how the loss of SMUG1 activity may promote cancer. Finally, we discuss recent studies indicating SMUG1 has moonlighting functions beyond BER, playing a critical role in RNA processing including the RNA component of telomerase.

## 1. Introduction

Damage to cellular DNA can be generated by both endogenous and exogenous sources, and life has evolved a series of intricate processes for the removal of these potentially mutagenic and carcinogenic injuries [1,2,3]. In mammalian cells, 1 of 11 DNA glycosylases initiate removal of base modifications resulting from alkylation, oxidation, and deamination during base excision repair (BER). While monofunctional glycosylases only contain glycosidic activity, resulting in abasic sites (apurinic/apyrimidinic), bifunctional glycosylases have a secondary enzymatic lyase activity, which breaks the sugar-phosphate backbone [4]. The resulting abasic site is further processed by an apurinic/apyrimidinic endonuclease 1 (APE1) generating a 3′OH to serve as a primer for DNA polymerase β (Pol β), which depending on the glycosylase, removes the 5′-deoxyribosephosphate (dRP) moiety and inserts the correct base. DNA ligase I or III seals the nick [5,6]. This review examines one specific glycosylase, single-strand selective monofunctional uracil DNA glycosylase (SMUG1), a protein from the uracil DNA glycosylase (UNG) family of glycosylases, which is responsible for the removal of uracil and oxidized pyrimidines from DNA. While we discuss this in more detail in a later section, it is important to clarify the name SMUG1 is a misnomer, as SMUG1 does not exclusively work on single stranded DNA [7,8,9]. 

One of the principal lesions SMUG1 targets is 5-hydroxymethyl-2′-deoxyuridine (5-hmdU), which has been measured at a frequency of 3.0/10^6^ bases in mammalian cells [10]. A second important SMUG1 substrate, which occurs at a lower frequency is the deamination product of cytosine, uracil, leading to G:U mismatches [11,12] (Figure 1). Unlike some glycosylases such as thymine DNA glycosylase (TDG), SMUG1 is not cell cycle regulated and is constitutively expressed [13]. In this review, we begin by describing the historical characterization of both 5-hmdU and SMUG1, focusing on the prominent structure-function features of SMUG1. We discuss the finding that SMUG1 is unable to work efficiently on lesions in the context of nucleosomes [14,15]. This review then describes the genomic consequences when SMUG1 expression and activity are lost, and these implications in cancer. We highlight recent studies indicating SMUG1 may play an important role in RNA processing and the maturation of the RNA component of telomerase. Lastly, we discuss the crosstalk which exists between DNA repair pathways in the context of oxidative DNA damage, and the role of SMUG1 in this process. 

## 2. 5-Hydroxymethyl-2′-Deoxyuridine (5-hmdU), the Target Substrate for SMUG1

The first evidence of 5-hmdU as a modification in DNA was from the work of the Marmur laboratory in the 1960s, which showed in bacteriophage DNA, thymine is replaced with a pyrimidine base, 5-hydroxymethyl uracil [16]. The authors postulated the bacteria used the synthesis of 5-hmdU to confer a metabolic advantage during infection. Over the course of the next decade, other groups [17,18] worked to chemically synthesize 5-hmdU. 

Understanding and identification of the sources of 5-hmdU lesions in mammalian cells began in the Goldstein group [19,20]. HeLa cells were incubated with radiolabeled [^3^H]-thymidine and were irradiated with ionizing radiation (IR). When the DNA was analyzed by HPLC, the authors detected the presence of 5-hmdU at a frequency of 1–1.5/10^5^ thymine modifications per 100 Gy of IR. Interestingly, the authors were unable to detect 5-hmdU in purified DNA that was treated with IR in solution. Characterization of 5-hmdU formation continued with work from Djurić et al., [21]. In this study, both immortalized breast epithelial cells (MCF-10A) and human breast cancer cells (MCF-7) were treated with hydrogen peroxide and 5-hmdU levels were quantified using gas chromatography and mass spectroscopy. The authors showed 5-hmdU levels in breast cancer cells were increased 2-fold from the base line levels of 1−2/10^4^ thymines after hydrogen peroxide treatment (10–200 uM), and 5-hmdU formation results from oxidative stress and not singlet oxygen, Figure 1.

More recently, a new source for 5-hmdU was uncovered: oxidation of thymine by ten-eleven translocation (TET) enzymes [10]. While the TET proteins are involved in the oxidative demethylation of 5-mC, several groups have shown that these enzymes work to oxidize thymine to 5-hmdU [10,22,23]. Pfaffeneder and co-workers utilized an isotope labeling strategy in combination with quantitative mass spectroscopy to determine the origin of 5-hmdU in mouse embryonic stem cells (mESCs) [24]. The authors were able to show the majority of 5-hmdU is generated as a result of thymine oxidation not cytosine deamination. Additionally, using shRNA to reduce Tet1 and Tet2 expression, decreased levels of 5-hmdU in the mESCs, and 5-hmdU levels were restored with ectopic expression of Tet1/2. 

The TET-induced formation of 5-hmdU was further investigated by Modrzejewska et al., in 2016 using ultraperformance liquid chromatography mass spectroscopy (UPLC-MS/MS) [25]. This precise labeling strategy was able to push the detection limits down to one 5-hmdU per 10^6^ deoxynucleotides (dN), comparable to levels of 8-oxoG. In this paper, the authors demonstrate high concentrations (1mM) of ascorbate lead to activation of TET, which could act as one potential source of 5-hmdU. In the presence of 1mM ascorbate, 5-hmdU levels increased to 20 per 10^6^ dN. The ascorbate dependent activation of TET has epigenetic implications, specifically in DNA demethylation which requires more investigation. 

## 3. Single-Strand-Selective Monofunctional Uracil-DNA Glycosylase 1 (SMUG1): A Short History 

### 3.1. Structure and Function of SMUG1

Identification of a 5-hmdU specific glycosylase began in the 1980s when Ames and Linn purified a protein that removes this moiety; it was named 5hmUra DNA N-glycosylase (HMUDG) [26]. Teebor and colleagues were then able to isolate HMUDG from calf thymus and demonstrated HMUDG was separate from that of uracil DNA glycosylase (UNG), by showing an increase in excision activity of 5-hmdU by this former enzyme [27]. The work from Cannon-Carlson and Teebor is one of the earliest implications for a distinction between the members of the UNG glycosylase family. Then in 1999, the Verdine group cloned and purified an activity, they called single-strand selective monofunctional uracil-DNA glycosylase (SMUG1). They employed a proteomics approach that utilized glycosylase inhibitors, to clone and characterize SMUG1 from both from *Xenopus* and human sources [28]. Kinetic analysis of the newly identified *Xenopus* protein revealed a preference for substrates within single stranded DNA. Lastly, the Verdine group used confocal microscopy to demonstrate SMUG1 localized in the nuclei of HeLa cells, consistent with the protein having a role in DNA repair. After the identification of SMUG1 by Haushalter et al., [28], the Teebor group returned to HMUDG and were able to specifically identify the protein as SMUG1 through mass spectroscopy and sequence alignment analysis [29]. The Lindahl laboratory, using knockout mice models, differentiated SMUG1 from UNG, by implying SMUG1 works outside of DNA replication [9]. Building upon the work from the Lindahl laboratory, work from the Slupphaug group demonstrated SMUG1 is not cell-cycle regulated, it is expressed constantly in cells, and predominantly localizes in nucleoli, where there are regions of both condensed and decondensed chromatin [13].

The co-crystal structure of *Xenopus* SMUG1 (xSMUG1) with a G:U mismatch was solved by the Verdine and Pearl laboratories at 2.0 Å resolution [30]. The structure indicated that SMUG1 belonged to the UDG family of glycosylases with a conserved α/β fold [8,30] containing the catalytic domain. The active site for xSMUG1 contains an asparagine residue (Asn85) in the N-terminus and a histidine residue (His239) in the C-terminus. There is a distinguishing structural feature within xSMUG1, a helical wedge that is composed of a five amino acid (251:P-S-P-R-N:255) loop and a five-residue (256:P-Q-A-N-K:260) α helix, which helps in damage recognition (Figure 2, highlighted in green). It is important to note the additional α helix found in the wedge is also only specific to xSMUG1. The authors proposed that xSMUG1 utilizes a water displacement strategy to identify target substrates (Figure 2B). Structurally, 5-hmdU differs from thymine by the hydroxyl group on the methyl moiety (Figure 1). In order to achieve uridine recognition xSMUG1 needs to form a bridging hydrogen bond with water and Gly98. Alternatively, xSMUG1 can displace the water and hydrogen bond with Met102 and Gly98 residues in hmU. The authors suggested that the methyl group in thymine prevents the water displacement mechanism from occurring, allowing for substrate differentiation. To further assess the mechanism of action of SMUG1, Matsubara et al., generated a variety of human SMUG1 (hSMUG1) mutants and performed enzyme activity assays to delineate key active site residues involved in damage recognition [7,31]. The mutagenesis studies revealed the key residues involved in the water displacement mechanism for damage recognition were Gly87 and Met91 (shown in red, Figure 2B), as well as highlighting the importance of the Asn85 and His239 residues in the hSMUG1 protein for glycosidic bond cleavage and Phe98 to differentiate the target base, see Figure 2B. 

### 3.2. SMUG1 Is Not Limited to Single-Stranded Substrates

As previously mentioned, the name of SMUG1 is a bit misleading as the activity of human SMUG1 is not limited to single stranded DNA. Further work by the Lindahl and Verdine laboratories demonstrated that SMUG1 is able to excise uracil from both single and double-stranded DNA with similar kinetics, but SMUG1 appeared to be product inhibited on double-stranded DNA. [9]. To this end, the authors sought to determine if the endonuclease, APE1 can facilitate SMUG1 turnover, and were able to demonstrate SMUG1 had enhanced activity on dsDNA in the presence of APE1 [9]. The product inhibition of SMUG1 was further investigated by [8] using an electrophoretic mobility shift assay (EMSA). The authors demonstrated strong binding (K_d_ = 0.125 ± 0.022 μM) of SMUG1 to abasic sites in double stranded DNA. Furthermore, SMUG1 in high amounts (over 1000-fold excess) was found to inhibit APE1 activity, suggesting that SMUG1 and APE1 compete for binding to an abasic site. In addition to 5-hmdU, SMUG1 can target other substrates including: uracil, 5-formyluracil, and 5-flurouracil [11]. Later work by Doseth et al. further demonstrated that mouse and human SMUG1 do not show preference for single stranded DNA, actually acting better on dsDNA substrates [32]. Interestingly, addition of both Mg^2+^ and APE1 increased uracil excision by SMUG1 2-fold. Moreover, the authors demonstrated the ability of SMUG1 to excise substrates from dsDNA using both isolated cell extracts and purified proteins. SMUG1’s substrate versatility extends past DNA. SMUG1 can also repair lesions found in RNA, and influence RNA quality control [33] which will be discussed in detail in the last section.

## 4. SMUG1′s Inability to Work on Substrates Embedded in Nucleosomes

Genomic DNA is organized into chromatin to help control transcription and preserve genomic integrity [34]. DNA, when not actively undergoing replication, is wrapped approximately twice around an octamer set of histones (two copies of H2A, H2B, H3, and H4) forming a nucleosome. The ability of SMUG1 to remove lesions in the context of nucleosomes was first studied by Nilsen, Lindahl, and Verreault using reconstituted nucleosome core particles (NCPs) containing U:A pairs [14]. They found that the efficiency of SMUG1 decreased approximately 9-fold when working on uracil embedded in nucleosomes (Figure 3). A subsequent study examined by the Delaney group examined the UNG glycosylase family, UNG, SMUG1, and TDG activities on NCPs containing a precisely positioned U:G wobble base pair as the targeted lesion [15]. The authors found SMUG1 activity was severely diminished for U:G wobble pairs regardless of the orientation on nucleosomes. Together these studies would suggest that SMUG1’s intercalating wedge is sterically hindered from accessing the DNA when it is wrapped around a nucleosome. This dramatic reduction in glycosylase activity of SMUG1 in the context of nucleosomes is intriguing and needs to be further investigated. It has been suggested that repair in chromatin proceeds through access-repair-restore model involving chromatin remodelers, histone chaperones, and histone variants [35,36], and it will be interesting to identify which factors help SMUG1 work on nucleosomes. 

## 5. SMUG1 Has Protective Roles in Cells

In bacteria, 5-hmdU was not believed to have cytotoxic effects. However, little was known about the effects of 5-hmdU on mammalian cells. The nucleoside, 5-hmdU can be phosphorylated and incorporated into DNA during replication. Recently, the thermodynamic implications of 5-hmdU on DNA polymerase activity and DNA replication were investigated by Hrabina et al., who demonstrated the presence of 5-hmdU in the DNA template does not lead to polymerase stalling [37]. Additionally, work from Litosh et. al demonstrated no change in PCR product formation when 5-hmdUTP is substituted for TTP [38]. One study indicated that despite the apparent lack of an effect on DNA replication, 5-hmdU is not mutagenic [39]. It is interesting to note that 5-hmdU incorporation into DNA can be toxic to mammalian cells. The Vilpo group was the first to demonstrate that 5-hmdU incorporation into human leukemia cell lines was cytotoxic [40]. Using increasing doses of 5-hmdU, they illustrated up to 50 percent reduction in cell proliferation after treatment. Additionally, using radiolabeled 5-hmdU the authors looked at net incorporation of the lesion into DNA and found there was very little being incorporated, implying 5-hmdU incorporation per se was not toxic, but it was the subsequent activity of repair enzymes which led to this toxicity. The nature of this toxicity may have to do with activation of PARP1 and subsequent bioenergetic collapse due to loss of NAD [41]; however, this model needs to be confirmed. 

Unlike SMUG1’s activity on 5-hmdU, which promotes cell killing, SMUG1 activity was found to be important in the protection from damage after ionizing radiation and fluorodeoxyuracil (Figure 4). An et al. used siRNA to knockdown SMUG1 in *Ung* knockout mouse embryonic fibroblasts (MEFs), to elucidate the role of SMUG1 cellular protection to ionizing radiation, as well as to characterize any functional overlap between SMUG1 and another UNG glycosylase, UNG [42]. The most potent killing effect by IR was shown in *Ung* KO MEFs depleted of SMUG1 with siRNA. Importantly the authors also studied the spontaneous mutation spectra in the *hprt* gene and were able to show a significant increase in C to T transition mutations in either *Ung*^−/−^ or SMUG1 knockdowns alone, and a synergistic increase of about 10-fold when both enzymes were lacking. The authors suggested this spontaneous mutation spectra was consistent with deamination at C residues, and they concluded that SMUG1 and UNG had non-overlapping functions in the removal of U:G mispairs. Another paper from the same group characterized the relationship between SMUG1 and a common cancer therapeutic, 5-fluoruracil (5-FU) [43]. Once again using *Ung* KO MEFs with siRNA mediated knockdown of SMUG1, they demonstrated a hypersensitivity to treatment with 5-FU. To this end, a correlative study was published and showed SMUG1 is upregulated in cells resistant to chemotherapy [44]. It would be interesting to further investigate the mechanism by which SMUG1 expression contributes to drug resistance. 

## 6. SMUG1 and Cancer

In order to better understand the dual roles of UNG and SMUG1 in genome stability, *Smug1^-/-^* mice were generated in 2012 and had no discernable phenotypic abnormalities showing SMUG1 is not critical for normal development [45]. Analysis of different tissues from the WT mouse showed that SMUG1 expression is highest in the brain. The authors also demonstrated that *Smug1^−/−^* MEFs lacked 5-hmdU excision activity, indicating SMUG1 is “the major if not sole enzyme” responsible for the removal of 5-hmdU [45]. Most importantly *Smug1^−/−^* MEFs showed marked resistant to 5-hmdU treatment, emphasizing the role for SMUG1 as the primary glycosylase for 5-hmdU lesions (Figure 5A). The nature of hmdU toxicity has not been well explored, but may be due to BER initiated PARP1 depletion of NAD [41]. Lastly, the authors were able to the show that mice carrying a loss of SMUG1 and UNG in combination with loss of mismatch repair (*Msh2^−/−^*) had shortened live spans and increased tumor formation. The shortened life span and increased tumor burden was significantly different in the *Smug1^−/−^ Ung^−/−^ Msh2^−/−^* mice versus *Ung^−/−^ Msh2^−/−^* mice implying a role for SMUG1 in protecting cells from genome instability induced cancer, perhaps through direct removal of hmdU moieties Figure 5B.

Based on these mice results, it might be expected that alterations in SMUG1 levels might be associated with human cancer (Figure 5C,D). To this end, high expression of SMUG1 is associated with favorable prognosis in ovarian cancer [47]. Additionally, low SMUG1 expression and mutations in SMUG1 are associated with poor prognosis in breast and colon cancer, respectively [48,49]. Abdel-Fatah and colleagues looked at SMUG1 mRNA or protein expression levels in triple negative breast cancer cells. In this study, low SMUG1 levels were associated with poor disease-free survival, increased histological grade characterization of tumors, enhanced cell proliferation, and an increase in BRCA1, ATM, or XRCC1 mutations. Mutations in SMUG1 have also been associated with poor prognosis in colon cancer [48,49]. In a study by Oliveira et al., [49] the authors analyzed the complete genome sequence on 37 patients. In addition to mutations in SMUG1, the genetic screen also found mutations in genes characteristically mutated in colon cancer including APC, KRAS, PIK3CA, and p53. Moreover, three patients with inactivating mutations in SMUG1 (Q93K, R187*, and R220W) demonstrated a 33% four-year survival rate after colon cancer diagnosis, compared to over 90% survival in patients without the mutations. It is important to note that these human studies comparing SMUG1 levels to cancer were only correlative. Thus, decreased SMUG1 activity and/or decreased expression in different forms of cancer remains largely uncharacterized on a molecular level. 

## 7. Moonlighting Functions of SMUG1 outside BER 

Besides providing an essential role in processing base damage during BER, evidence is growing that SMUG1 plays important roles in RNA maturation and RNA quality control. SMUG1 has been found to localize in the nucleoli, and Jobert and colleagues illustrated co-localization of SMUG1 with Cajal bodies [33]. Furthermore, using immunofluorescence (IF) and immunoprecipitation (IP) experiments the authors demonstrated an association between SMUG1 and Dyskerin (DKC1), a protein involved in RNA processing. They found evidence that these two proteins working together helped to remove and degrade damaged RNA from the cell. An IP was performed to confirm an interaction between SMUG1 and DKC1. This interaction was observed in both the presence and absence of RNAse and DNase, implying a direct protein–protein interaction. In order to test the hypothesis that SMUG1 may be involved in RNA damage detection and removal, the authors examined the activity of SMUG1 on RNA containing 5-hmdU. Incision assays using a defined RNA substate containing either uridine, pseudouridine, or 5-hmdU lesions demonstrated that SMUG1 had the highest activity for 5-hmdU. Additionally, the authors generated a catalytically dead SMUG1 mutant which was capable of binding to lesions but displayed loss of excision activity of 5-hmdU from RNA. The authors further explored the binding between SMUG1 and RNA by performing a co-IP to determine if SMUG1 interacts with specific subunits of ribosomal RNA (rRNA). After co-immunoprecipitating RNA and SMUG1, the authors utilized reverse-transcription quantitative PCR (RT-qPCR) with primers specific to various subunits of rRNA including: 47S, 28S, 18S, and 5.8S. The RT-qPCR revealed SMUG1 binds strongly to the 47S subunit of rRNA, which is a target of DKC1. In addition, the authors utilized siRNA mediated knockdowns of SMUG1 to show loss of SMUG1 expression leads to a decrease in processed rRNA subunits, including the 28S, 18S, and 5.8S. However, the levels of the 47S subunit remained intact after SMUG1 knockdown. Together these data suggest without SMUG1, unprocessed rRNA accumulates. Moreover, the authors examined 5-hmdU accumulation using liquid chromatography-tandem mass spectrometry (LC-MS/MS) in different rRNA molecules after knocking down SMUG1 expression and were able to demonstrate an increase in 5-hmdU levels. Lastly, the authors used the SMUG1 mutant to demonstrate SMUG1 needed to be catalytically active to interact with DKC1 and to coordinate proper nuclear localization. 

In addition to having a role in RNA processing, DKC1 is a component of the dyskerin complex within the telomerase holoenzyme. Telomerase works at the ends of telomeres to add repeated sequences to promote lengthening of chromosome ends [50]. Telomerase has three main components: the dyskerin complex, the telomerase reverse transcriptase (hTERT), and the telomerase RNA component (hTERC). After establishing an interaction between SMUG1 and DKC1, the Nilson laboratory wanted to determine if SMUG1 had a greater role in proper telomere development [51]. Using the telomere chromatin immunoprecipitation (TeloCHIP) assay, the authors were able to show an interaction between SMUG1 and DNA at the telomeres, and subsequently the interaction was lost in *Smug1*^−/−^ mice. Additionally, in *Smug1*^−/−^ mice the authors demonstrated an increase in defective telomeres as illustrated by a reduction in qPCR efficiency, emphasizing an increase in damaged bases which decreases amplification of telomeric DNA. The *Smug1*^−/−^ mice also presented with a decrease in telomere length. After identifying the telomere defects, the authors examined whether telomerase activity was altered in *Smug1*^−/−^ mice. To this end the authors demonstrated a decrease in telomerase activity in the *Smug1*^−/−^ mice and furthermore a decrease in hTERC expression demonstrating a role for SMUG1 in proper hTERC development. Using an RNA-IP the authors demonstrated without SMUG1, DKC1, and hTERC are unable to properly associate, leading to an increase in immature hTERC protein complexes which contribute to the decrease in telomerase activity. Additionally, the authors demonstrated the glycosylase activity of SMUG1 had a role in telomerase maintenance by showing SMUG1 was only able to bind hTERC if there was a damage base present. Both of these studies by the Nilsen group illustrate roles for SMUG1 outside of base damage repair and illustrates the need to further characterize how SMUG1 may provide other important cellular functions.

### XPC and CSB Stimulate SMUG1

While it was previously believed oxidative base damage is solely remedied through base excision repair, there is growing evidence of crosstalk between BER and another DNA repair pathway, nucleotide excision repair (NER). The substrate repertoire for NER is more diverse than that of BER, as NER can repair a wide range of lesions, including UV-induced photoproducts and other bulky carcinogenic adducts [52]. Nucleotide excision repair can be further subcategorized into two pathways: global genome NER (GG-NER) and transcription-coupled NER (TC-NER), which are distinguished based on how the lesions are initially recognized [52]. The crosstalk between NER protein and BER proteins in the removal of oxidative damage has been reviewed [53,54,55]. The role for a GG-NER recognition protein, XPC, in stimulation of SMUG1 activity was demonstrated by the Sugasawa group [56]. Using enzyme activity assays containing a G:U mismatch or a single stranded oligo with a uracil as the substrate, SMUG1 cleavage activity was measured in the presence of increasing concentrations of XPC-RAD23B. Interestingly, the authors revealed that XPC stimulated SMUG1 four-fold. Furthermore, the authors performed a GST-pulldown assay using SMUG1 and XPC to confirm a direct interaction between these two proteins. The crosstalk can also be extended to TC-NER. Wong et al., demonstrated hypersensitivity, approximately three-fold, to 5-hmdU treatment in cells with mutated CSB protein [57]. This finding, which suggests an involvement of CSB in SMUG1 driven removal of 5-hmdU, needs further investigation. 

## 8. Conclusions and Outlook 

Base modifications contribute greatly to genomic instability and therefore there is a great need to understand the repair mechanisms dedicated to damage removal. In this review, we discussed in detail one key protein involved in base excision repair, SMUG1. Outside of uracil, SMUG1 primarily works on 5-hmdU, a lesion which has been demonstrated to be elevated in tumor cells [21]. SMUG1 is a part of the uracil-DNA glycosylase family, which is structurally defined by an α/β motif with a positively charged groove which facilitates binding between the DNA and the glycosylase [58]. SMUG1 has a unique structural characteristic of a helical wedge, which is utilized for substrate recognition [30]. The wedge helps SMUG1 to distort the DNA helix to increase lesion accessibility. When SMUG1 expression is lost, cells are increasingly sensitive to the effects of IR and the chemotherapeutic agent, 5-FU, but become resistant to 5-hmdU treatment. Some studies suggest a loss of SMUG1 increases susceptible to cancer development [42,43,45].

5-hmdU is incorporated into the DNA during replication and has little to no detrimental effect on DNA polymerases [37,38,39]. One unresolved question is whether 5-hmdU toxicity is due to downstream events in BER. Furthermore, it is not known whether telomerase is capable of inserting 5-hmdU. Previous work looked at understanding the effect of oxidized guanine damage in the context of telomeres [59]. Oxidative base damage prevents proper formation of the shelterin complex, inducing a DNA damage response leading to the formation of telomere dysfunction-induced foci (TIFs). However, it is still unknown the impact of oxidized thymine moiety and pyrimidine base damage on telomere stability and maintenance. It would be of note to investigate if after telomerase mediated incorporation of 5-hmdU, there is an increase in TIFs and subsequently, SMUG1 recruitment to telomeres.

We discussed the importance of nucleosomes and chromatin compaction in genomic stability. Remarkably, SMUG1 activity is significantly reduced when the target lesion is embedded within a nucleosome [15]. Lesion accessibility is a contributing factor in the progression of BER, and as previously mentioned loss of BER can lead to cancer development. Therefore, a great need exists to understand why SMUG1 does not function efficiently on lesions in the context of nucleosomes and if there are ways to overcome this inhibition. Chromatin decompaction is an important aspect of several DNA repair pathways. To this end, our laboratory recently described a role for UV-damaged DNA binding protein (UV-DDB) in the BER of 8-oxoguanine [60]. UV-DDB is a heterodimeric protein, consisting of DDB1 and DDB2 subunits, and acts as first responder to UV-damage during nucleotide excision repair (NER) [61]. After UV-damage, UV-DDB forms an E3-ubiqutin ligase complex with Cul4A and RBX to ubiquitinate histones and remodel the chromatin to increase lesion accessibility for downstream repair proteins [62]. Recently, our laboratory demonstrated a non-canonical role for UV-DDB in BER, by showing UV-DDB can stimulate the activity of the glycosylase, 8-oxoG glycosylase (OGG1) and the endonuclease APE1, three-fold and eight-fold, respectively. SMUG1 is product inhibited as it binds more avidly to abasic sites than 5-hmdU, indicating a need for other proteins to facilitate glycosylase turnover to all progression and completion of repair. To this end, current work in our laboratory is focusing on UV-DDB mediated stimulation of the other 10 mammalian glycosylases, including SMUG1. We have preliminary data suggesting UV-DDB can stimulate SMUG1 activity. The mechanisms by which SMUG1 works both biochemically and in the cell remain largely uncharacterized. Our work reiterates the point there is significant crosstalk amongst DNA repair pathways which need to be further investigated. 

## Figures and Tables

**Figure 1 ijms-22-01981-f001:**
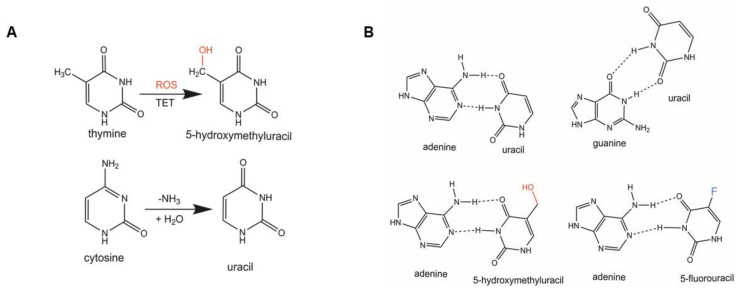
Generation of 5-hmdU and structures of pertinent SMUG1-targeted oxidative lesions. Chemical structures of SMUG1 lesions. (**A**) Formation of pertinent SMUG1 lesions: 5-hydroxymethyluracil (5-hmU) after oxidation of thymine by reactive oxygen species (ROS) or ten-eleven translocation (TET) enzymes. (**B**) Base pairings of SMUG1 target substrates. The hydroxyl moiety generated by ROS is highlighted in red.

**Figure 2 ijms-22-01981-f002:**
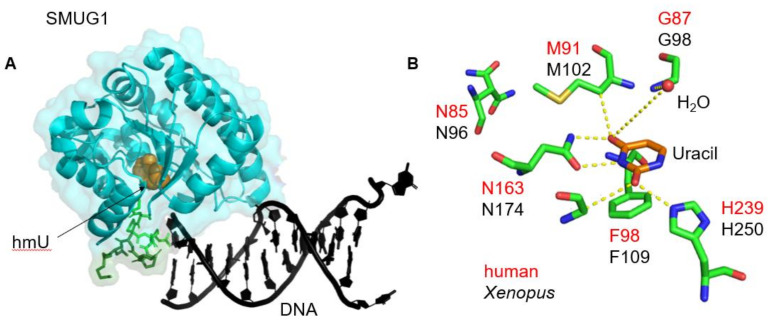
Structure of SMUG1 highlighting the role of the wedge domain in catalysis. Crystal structure of Xenopus SMUG1 (**A**) The glycosylase (blue) bound to DNA (black) containing an abasic site with the damage recognition wedge (loop, light green; helix, dark green) with the free 5-hmdU (orange), PDB: 1OE6. (**B**) Key active site residues of SMUG1 (PDB: 1OE5), with Xenopus (black) and human (red) numberings; adapted from [8] with permission.

**Figure 3 ijms-22-01981-f003:**
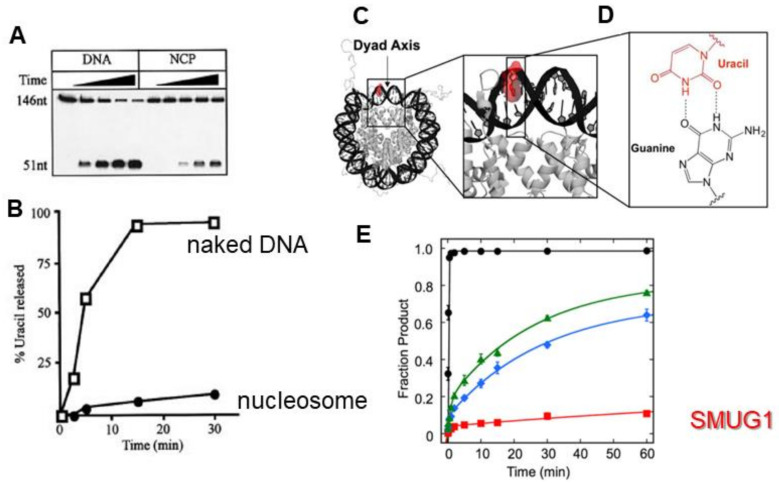
SMUG1 processes lesions within nucleosomes inefficiently. SMUG1 (**A**) excision activity and (**B**) kinetics on naked DNA and nucleosome core particle (NCP), from [14] with permissions. (**C**) Crystal structure of the NCP highlighting the (**D**) U:G wobble base pair lesion from [15] with permission. (**E**) SMUG1 excision kinetics (red) compared to the activity of other UDG glycosylases, UNG (black) TDG (blue, green) from [15] with permission.

**Figure 4 ijms-22-01981-f004:**
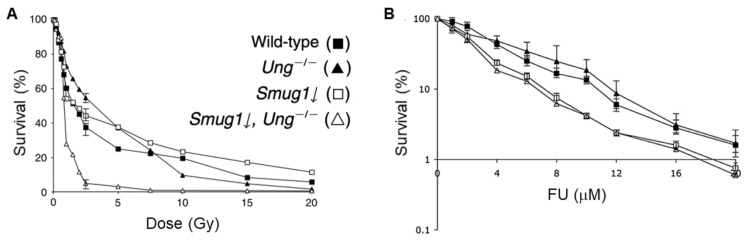
SMUG1 has a protective role after specific types of DNA damage. Survival of wildtype, Ung-/-,Smug1 knockdown and Smug1 knockdown and Ung^−/−^ MEFs were analyzed after treatment with: (**A**) ionizing radiation, from [42] with permission, or (**B**) 5-fluorouracil (FU) from [43] with permission.

**Figure 5 ijms-22-01981-f005:**
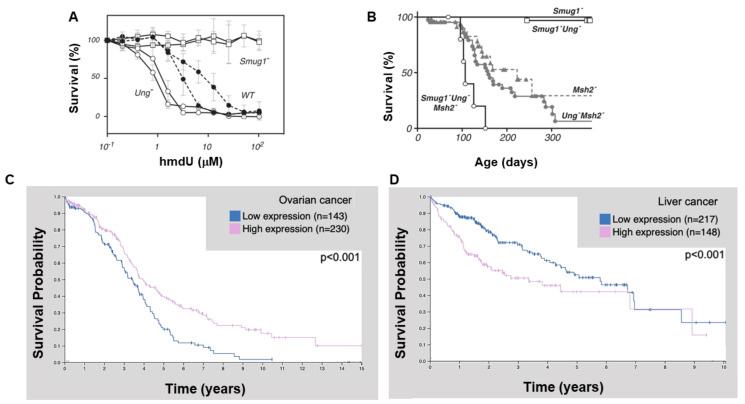
SMUG1 and cancer development and progression. The relationship between SMUG1 expression and cancer progression. (**A**) Effects of Smug1^−/−^ Ung^−/−^ Msh2^−/−^ on mice survival from [45] with permission. (**B**) Smug1^−/−^ has protective effect to treatment with 5-hmdU, from with [45] permission. Summary of the search of The Human Protein Atlas for SMUG1 in the pathology section. Prognostic analysis of patient survival in (**C**) ovarian and (**D**) liver cancers. Low mRNA expression (blue) compared to high (pink) mRNA expression levels of SMUG1, analysis reached high significance [46].

## Data Availability

Not applicable.

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
