# Peer review of "The Multiple Cellular Roles of SMUG1 in Genome Maintenance and Cancer"

_ijms, 2021, doi:10.3390/ijms22041981_

Round 1

Reviewer 1 Report

This is a well-written review article on the biological functions of SMUG1 in BER, NER and RNA processing and repair. In addition, the authors also briefly discussed the correlation between the expression level of SMUG1 and the prognosis of different cancers as well as their response to chemotherapy. The authors also provided appropriate figures to make the reading more pleasant. I strongly recommend its acceptance for publication.

Author Response

We thank Reviewer 1 for their nice feedback and enthusiasm for the manuscript.

Reviewer 2 Report

This is an interesting review on the multiple cellular roles of SMUG1 in genome maintenance and cancer. I have few comments on the article.

  1. The authors mention that NER protein XPC was involved in stimulation of SMUG1 activity. Is there any report on the involvement of XPA in this process.
  2. Does SMUG1 regulates immune response?
  3. Does SMUG1 affect the epigenome in addition to the genome?

Author Response

  1. This is a great question. From our current review of the literature it is unclear if there are other NER proteins, such as XPC or XPA, involved in the stimulation of SMUG1. This is an area which needs further study.
  2. Uracil has been shown to have a role in immunoglobin (Ig) development, and this was investigated by (Doseth B, Visnes T, Wallenius A, et al. Uracil-DNA glycosylase in base excision repair and adaptive immunity: species differences between man and mouse. J Biol Chem. 2011;286(19):16669-16680. doi:10.1074/jbc.M111.230052), who demonstrated SMUG1 activity increased after B-cell activation. However from our current understandings of the literature the role of SMUG1 in immune response regulation needs to be further investigated.
  3. The epigenetic implications of SMUG1 have not been widely characterized presently, but given the role of TET enzymes in 5-hmdU formation SMUG1 could be regulated by epigenetic modifications.

Reviewer 3 Report

Raja  et Van Houten  present a review on multiple cellular roles of SMUG1.

The topic is presented in a clear and exhaustive way. The authors have described in a simple way the latest findings regarding the role and function of SMUG1 in recognizing mismatch in single / double strand DNA and also its repairing action of the hmU lesion on RNA. 

The text is well written and very clear. There is a typographical error on line 11.

Author Response

We thank reviewer 3 for their nice feedback and appreciate them catching typographical error in line 11 of the text and this has since been corrected.